



# Towards operational phytoplankton recognition with automated high-throughput imaging and compact convolutional neural networks

Tuomas Eerola[1], Kaisa Kraft[2], Osku Grönberg[1], Lasse Lensu[1], Sanna Suikkanen[2], Jukka Seppälä[2], Timo Tamminen[2], Heikki Kälviäinen[1], and Heikki Haario[1]

[1]Computer Vision and Pattern Recognition Laboratory, School of Engineering Science, Lappeenranta-Lahti University of Technology LUT, Finland, *firstname.lastname@lut.fi*
[2]Finnish Environment Institute, Marine Research Centre, Helsinki, Finland, *firstname.lastname@ymparisto.fi*

**Correspondence:** Tuomas Eerola (tuomas.eerola@lut.fi)

**Abstract.** Plankton communities form the basis of aquatic ecosystems and elucidating their role in increasingly important environmental issues is a constantly present research question. The concealed plankton community dynamics reflect changes in environmental forcing, growth traits of competing species, and multiple food web interactions. Recent technological advances have led to the possibility of collecting real-time big data opening new horizons for testing core hypotheses in planktonic sys-

tems, derived from macroscopic realms, in community ecology, biodiversity research, and ecosystem functioning. Analyzing the big data calls for computer vision and machine learning methods capable of producing interoperable data across platforms and systems. In this paper we apply convolutional neural networks (CNN) to classify a brackish-water phytoplankton community in the Baltic Sea. For solving the classification task, we utilize compact CNN architectures requiring less computational capacity and creating an opportunity to quickly train the network. This makes it possible to 1) test various modifications to

the classification method, and 2) repeat each experiment multiple times with different training and test set combinations to obtain reliable results. We further analyze the effect of large class imbalance to the CNN performance, and test relevant data augmentation techniques to improve the performance. Finally, we address the practical implications of the classification performance to aquatic research by analyzing the confused classes and their effect on the reliability of the automatic plankton recognition system, to guide further development of plankton recognition research. Our results show that it is possible to ob-

tain good classification accuracy with relatively shallow architectures and a small amount of training data when using effective data augmentation methods even with a very unbalanced dataset.

## 1 Introduction

Research on plankton communities is hampered by the bottleneck of acquiring species-level information on these small-sized organisms, requiring laborious sample preparation and microscopic identification. Globally the key primary producers, the

phytoplankton communities, consist of hundreds of microorganism species with generation times down to the order of hours. Zooplankton, another important component of planktonic food webs, are roughly an order of magnitude smaller in species



numbers, and their larger size also makes species identification somewhat easier. Nevertheless, planktonic systems as a whole remain a methodological challenge for elucidating their role in increasingly important environmental issues: the concealed plankton community dynamics reflect changes in environmental forcing, growth traits of competing species, and multiple food
web interactions. Recent technological advances have led to the emergence of automated and semi-automated imaging instruments, with steadily improving image resolution and output rates, up to tens of thousands of images per hour (e.g. Lombard et al., 2019). It is already possible to produce real-time Big Data of plankton communities. This opens new horizons for testing core hypotheses in planktonic systems, derived from macroscopic realms, in community ecology, biodiversity research, and ecosystem functioning.

However, the next bottleneck is obvious. No human can ever screen the millions of images. Analyzing the big data calls for computer vision and machine learning methods capable of producing interoperable data across platforms and systems. To enable the automatic analysis, the traditional approach for automatic plankton classification starts with the extraction of manually-engineered image features such as shape and texture. The features are then used to train a classifier, typically either a Support Vector Machine (SVM) (Cortes and Vapnik, 1995) or a Random Decision Forest (RDF) (Ho, 1995). The trained
classifier can be applied to the features extracted from new images to recognize the plankton species (see e.g. Bueno et al., 2017).

The main problem in the traditional approach is to find image features that are both general and provide good discrimination between the classes. This is not an easy task and often leads to suboptimal classification accuracy and poor generalization to other domains, i.e., the developed methods tend to only work on the set of plankton species and imaging method they were
developed with. However, recent progress in both computer vision techniques and computing resources has made it possible to learn suitable image features from data (LeCun et al., 2015).

Recent advances in utilizing deep learning techniques for phytoplankton identification, especially Convolutional Neural Networks (CNNs), have shown them as an attractive choice for automating the process (e.g. Lumini and Nanni, 2019; Luo et al., 2018). CNNs can be efficiently used to learn the image features relevant for the identification task without the need for
manual engineering. Given a set of example images from each output class (training data), a CNN simultaneously learns in the supervised manner multiple filter layers formulated as sets of computational neurons performing convolutions and fully connected classification layers in such way that the classification error in the training data is minimized. CNNs have been shown to achieve state-of-the-art performance in various image analysis tasks including image classification (He et al., 2016) and object detection (Ren et al., 2015). There have also been several studies on applying CNN-based classification methods
successfully on plankton images.

Classification of mesoscale plankton has been a more approachable task than that of microplankton, due to the limited optical resolution of imaging devices. Species with larger sizes tend to have more distinguishable features between different groups and with the technology available the overall collection of image datasets has been easier. Many studies have focused on the image classification task utilizing open-access datasets collected with the laboratory-based mesoplankton imaging system ZooScan,
and the in-situ mesoplankton imaging systems ISIIS and SIPPER (e.g. Al-Barazanchi et al., 2015; Dai et al., 2016a; Li and Cui, 2016). An especially popular image set has been the one used in the 2015 National Data Science Bowl in Kaggle (Cowen



et al., 2015), which exhilarated the utilization of CNNs for plankton classification. Along with the technical development, imaging of smaller organisms has become easier with improved image resolution and quality. Several imaging systems have become commercially available, and are now used more frequently both for science and monitoring purposes, making it easier
to collect image datasets (Lombard et al., 2019).

One of the most promising methods for observing smaller-scale aquatic organisms is imaging flow cytometry. Imaging FlowCytobot (IFCB) (Olson and Sosik, 2007) is among the most frequently used imaging flow cytometers and there are promising results of its utilization in phytoplankton ecology studies (e.g. Laney and Sosik, 2014; Harred and Campbell, 2014; Anglès et al., 2019). Recent studies have shown CNNs to perform well also when classifying smaller-sized plankton, image
datasets comprising nanophytoplankton, microphytoplankton and other microplankton such as ciliates (Lumini and Nanni, 2019; Orenstein and Beijbom, 2017; Dai et al., 2016b; Lee et al., 2016; Li and Cui, 2016). The studies mentioned above have used the most popular plankton image dataset available, WHOI-Plankton, which consists of millions of images captured with an IFCB at Martha's Vineyard Coastal Observatory of the Woods Hole Oceanographic Institution (WHOI), US (Orenstein et al., 2015). CNN-based methods have also been successfully applied to image sets collected with imaging systems other than
the IFCB, such as the imaging flow cytometer FlowCAM (Correa et al., 2017), a microscopy-based system (Li et al., 2019) and imaging systems resembling more traditional flow cytometers such as Image Stream (Dunker et al., 2018).

Multiple deep learning platforms with pre-trained generic CNN models are available (e.g. TensorFlow, Keras, PyTorch), and consequently, choices for CNN architectures and training procedures have been numerous in recent publications. However, applying CNN techniques to an image recognition problem is not straightforward due to non-ideal distribution of data and
differences in design choices for CNN architecture. Our approach in this study is to address some fundamental challenges in phytoplankton identification as a first step, and at the same time, initiate compilation of an annotated data set for further image recognition studies from phytoplankton communities of the Baltic Sea. This environment is one of the Earth's largest brackish water habitats, with an especially challenging mix of phytoplankton species of both freshwater and marine origin (Olli et al., 2019).

Typically, papers on plankton classification only report the classification performance and do not account for the practical implications to aquatic research. We address this by analyzing the confused classes and their effect on the reliability of the automatic plankton recognition system, to guide further development of plankton recognition research. We analyze how large class imbalance (from ten to thousands of example images per class) affects the CNN performance, and test relevant data augmentation techniques to improve the performance. CNN-based methods are usually trained using datasets with hundreds, or
even thousands of example images from each class, which is often difficult to obtain in practice in new locations. In the process of making the step towards automated/semi-automated plankton classification for real-time plankton observations, also the size of the CNN architecture used becomes an important factor determining the computational capacity needed. Therefore, we utilize relatively compact architectures to be able to quickly train the network, making it possible to 1) test various modifications to the classification method, and 2) repeat each experiment multiple times with different training and test set combinations to
obtain reliable results.





High-throughput imaging coupled with efficient deep learning techniques is one of key game changers in ecological phytoplankton research. As with many other branches of science utilizing big data, key challenges in plankton imaging are related to validation of data quality, integration of different data sources, common vocabularies of metadata, sharing of data and technology solutions and how to create reliable, acceptable and timely products. In their review, Lombard et al. (2019) list a set

of challenges and priorities for emerging phytoplankton detection technologies and one of their main recommendations is collaboration between experts and exchange with other disciplines, like modelers. Phytoplankton imaging is recognized as one of the main emerging technologies also within the pan-European coastal observing JERICO-Research Infrastructure consortium (Puillat et al., 2016; Farcy et al., 2019), and forming an important part of the future European integrated coastal observation system. Our study is one step in the technology development towards these goals, getting the phytoplankton recognition using

imaging systems more reliable.

## 2 Materials and methods

The data consist of images of phytoplankton captured with the IFCB (Olson and Sosik, 2007). The IFCB is an *in situ* automated submersible imaging flow cytometer. It captures images of suspended particles in the size range of 10 to 150 $\mu$m, with an image resolution of roughly 3.4 pixels per $\mu$m. The device samples seawater at a rate of 15 ml per hour and can produce

tens of thousands of images per hour, depending on the cell densities. The IFCB also gives analog-to-digital converted data from the photomultiplier tubes of the device. The photomultiplier tubes are used to detect light scatter and fluorescence from particles excited by the device laser and the analog-to-digital converted values of the photomultiplier tubes are used to determine whether an image should be captured or not. The data was collected from the northern Baltic Sea in autumn 2016 and from spring to summer 2017 (Fig. 1). The 2016 data was collected from the Alg@line ferrybox system of M/S Finnmaid and Silja

Serenade (Ruokanen et al., 2003; Kaitala et al., 2014). The 2017 data originate from the Utö Atmospheric and Marine Research Station (Laakso et al., 2018). Example images of the data set can be seen in Fig. 2.

The dataset contains grayscale images and metadata of phytoplankton divided into 53 different classes (not representing taxonomic phytoplankton classes). Furthermore, the following classes were divided into subclasses based on visual characteristics: Ciliata, *Chroococcus*, Cryptophyceae, Dinophyceae, *Mesodinium rubrum*, Pennales, and *Snowella/Woronichinia* sp.

The reason for this was the assumption that dividing classes with two or more visually different clusters into subclasses would improve the classification result. The total amount of classes with the subclasses included was 61.

There is large variation in the sizes and aspect ratios of the images. The vertical axes of the images range from 21 pixels to 770 pixels and the horizontal axes of the images from 52 pixels to 1,359 pixels. There is also a very large imbalance in the number of images per class as shown in Table 1. The number of images per class varies from 10 (*Apedinella radians*) to 3710

(*Snowella/Woronichinia* sp.).

The metadata used in the experiments contain the original image dimensions and analog-to-digital converted data of the average and peak values of the photomultiplier tubes during a laser pulse. The metadata was pseudo-normalized to have values between -1 and 1. Four data subsets named Subset100, Subset50, Subset20, and Subset10 were constructed from the data by



**Table 1.** Classes, their taxonomic group, and the number of images per class in the dataset.

| Class label | Taxonomic group | Images | Class label | Taxonomic group | Images |
|---|---|---|---|---|---|
| *Snowella / Woronichinia* sp. | Cyanobacteria | 3710 | *Merismopedia* sp. | Cyanobacteria | 138 |
| Dinophyceae | Dinoflagellates | 2268 | *Gymnodinium* sp. | Dinoflagellates | 133 |
| Chroococcales | Cyanobacteria | 1553 | Calibration bead | | 100 |
| *Heterocapsa triquetra* | Dinoflagellates | 1433 | Chlorococcales | Chlorophytes | 88 |
| *Dolichospermum / Anabaenopsis* sp. | Cyanobacteria | 1223 | *Katablepharis remigera* | | 85 |
| *Chaetoceros* sp. | Diatoms | 916 | *Nodularia spumigena* | Cyanobacteria | 80 |
| *Peridiniella catenata* single cell | Dinoflagellates | 871 | *Dinophysis acuminata* | Dinoflagellates | 73 |
| *Pseudopedinella* sp. | Chrysophytes | 829 | Unidentified species 1 | | 72 |
| *Aphanizomenon flosaquae* | Cyanobacteria | 821 | *Uroglenopsis* sp. | Chrysophytes | 69 |
| *Skeletonema marinoi* | Diatoms | 756 | *Licmophora* sp. | Diatoms | 62 |
| *Thalassiosira levanderi* | Diatoms | 650 | *Cyclotella choctawhatcheeana* | Diatoms | 55 |
| *Mesodinium rubrum* | Ciliates | 633 | Euglenophyceae | Euglenophytes | 42 |
| *Pyramimonas* sp. | Chlorophytes | 623 | *Ceratoneis closterium* | Diatoms | 33 |
| *Heterocapsa rotundata* | Dinoflagellates | 569 | Gymnodiniales | Dinoflagellates | 31 |
| *Oocystis* sp. | Chlorophytes | 458 | *Aphanothece paralleliformis* | Cyanobacteria | 29 |
| *Teleaulax* sp. | Cryptophytes | 413 | *Chaetoceros similis* | Diatoms | 24 |
| Centrales | Diatoms | 249 | *Melosira arctica* | Diatoms | 24 |
| *Prorocentrum cordatum* | Dinoflagellates | 230 | Akinete | Cyanobacteria | 23 |
| Heterocyte | Cyanobacteria | 225 | *Amylax triacantha* | Dinoflagellates | 21 |
| Ciliata | Ciliates | 201 | *Monoraphidium contortum* | Chlorophytes | 19 |
| Pennales | Diatoms | 198 | Oscillatoriales | Cyanobacteria | 15 |
| *Eutreptiella* sp. | Euglenophytes | 190 | *Binuclearia lauterbornii* | Chlorophytes | 13 |
| Cryptophyceae | Cryptophytes | 175 | *Pauliella taeniata* | Diatoms | 13 |
| *Cymbomonas tetramitiformis* | Chlorophytes | 150 | *Scenedesmus* sp. | Chlorophytes | 13 |
| Cyst | Dinoflagellates | 150 | *Chaetoceros throndsenii* | Diatoms | 12 |
| *Peridiniella catenata* chain | Dinoflagellates | 140 | *Apedinella radians* | Chrysophytes | 10 |
| Cryptomonadales | Cryptophytes | 138 | | | |


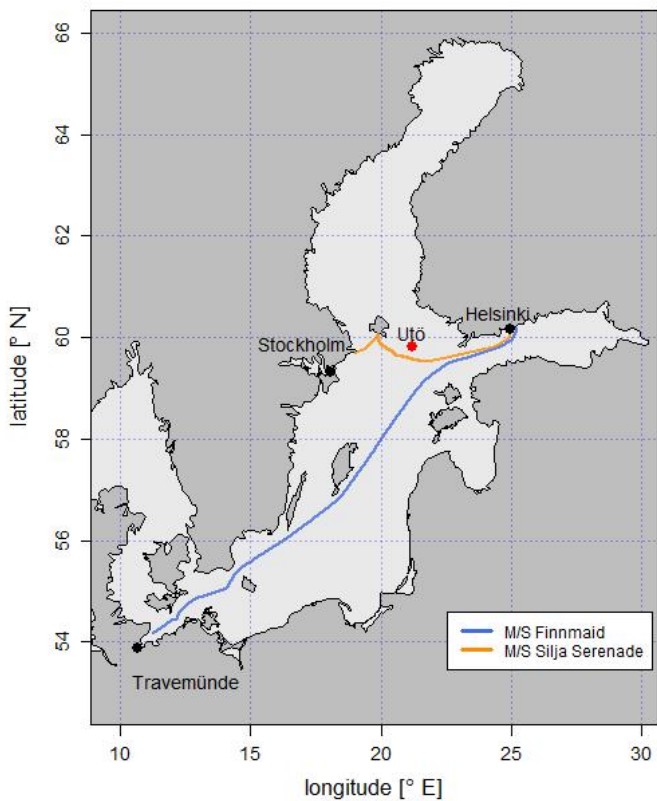

**Figure 1.** Map of the Baltic Sea with Utö island and the Alg@line routes of M/S Finnmaid and M/S Silja Serenade, from where the image data has been collected.

selecting classes with the number of images equal to or greater than the selected threshold value. For example, Subset100

contains all the classes in the data with at least 100 images per class. The details of these data subsets are shown in Table 2. Each data subset was randomly split into training sets and testing sets. The number of images in a subset assigned to the testing set is equal to 25% of the threshold value of the subset. The remaining images, up to one thousand images, are then assigned to the training set.

## 2.1 Data preprocessing

A CNN typically requires the input images to have a constant and predefined size. Resizing plankton images to a uniform size was done by reducing the size of the large images with bicubic interpolation so that the larger axis of the image matches the




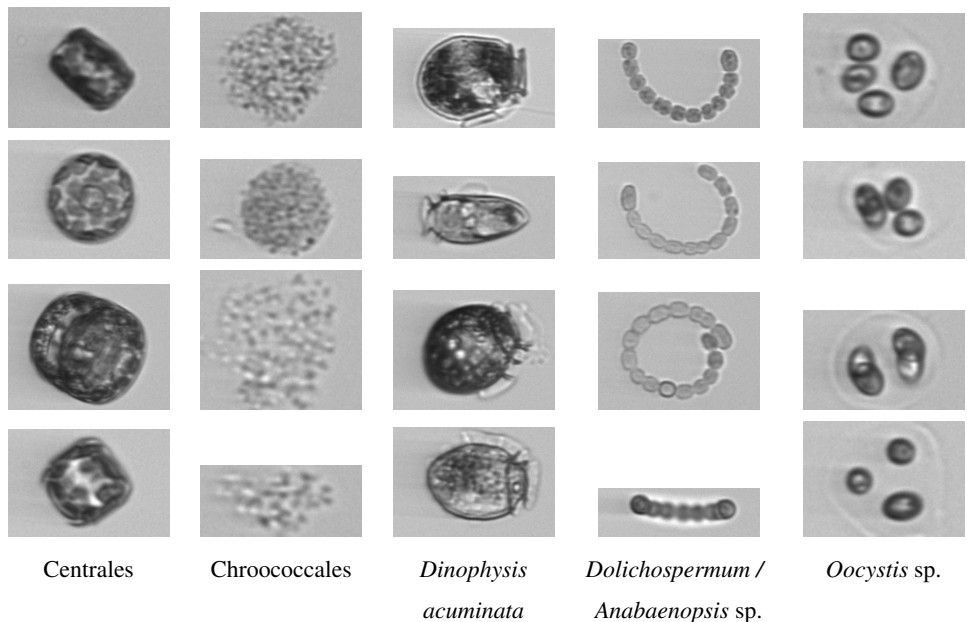

| Centrales | Chroococcales | *Dinophysis acuminata* | *Dolichospermum / Anabaenopsis* sp. | *Oocystis* sp. |

**Figure 2.** Example phytoplankton images from the dataset.

**Table 2.** Data subsets for the experiments.

| Name | Threshold | Number of classes (including subclasses) | Minimum number of training images per class | Test images per class |
|---|---|---|---|---|
| Subset100 | Classes with ≥100 images | 34 | 75 | 25 |
| Subset50 | Classes with ≥50 images | 43 | 38 | 12 |
| Subset20 | Classes with ≥20 images | 54 | 15 | 5 |
| Subset10 | Classes with ≥10 images | 61 | 8 | 2 |

target frame. The target frame refers to the desired size of the resized images and the image aspect ratio was kept as constant as possible. Then the image was padded to match the target frame dimensions. Images that are smaller than the target frame were enlarged to have their larger axis match the target frame and then the images were padded to fit the target frame. As there is a lot of background in the images, the mode of the image, that is, the most common pixel value in the image, was used as the padding color for the image.

Data augmentation was used to increase the number of training images. The implemented data augmentation methods include reflection, 90-degree rotation and cropping 10 percent off both ends of the larger image axis of the original training images. The cropping was performed before the images were resized to the uniform size. By using the augmentation methods, adding the reflected, cropped and rotated versions of the training set images increased the amount of training data by a factor of two, two and four, respectively. Therefore, the proposed augmentation allowed the generation of 16 times more training data for classes





with few training images. Because of the small number of images in a number of classes, data augmentation was performed on the training sets until each class contained one thousand images. Duplicates of the images were added if there were less than one thousand images in a class after the data augmentation.

## 145  2.2  Proposed CNN architectures

The proposed network architectures are based on AlexNet (Krizhevsky et al., 2012). Validating any results related to class-specific classification accuracies becomes computationally expensive since there are few images in a moderate number of classes. As a result, more compact network architectures are more ideal for studying the impact of different designs and their effect on the phytoplankton classification accuracy. The sequential network structures of the proposed networks can be seen in
Tables 3 and 4, where $n$ is the number of classes in data, $p$ is the probability to dropout an activation in the network and $M$ is the number of metadata features passed to the network to give additional inputs to the model. The feature vector passed as the secondary input is concatenated with the flattened feature map of the last convolutional layer.

**Table 3.** The sequential structure of CNN128. The network consists of roughly three million trainable parameters. The number of parameters depends on the number of classes and the number of features given form the secondary input.

| Layer | Layer dimensions | Parameters | Output dimensions |
|---|---|---|---|
| input | 128x128 | | 128x128 |
| convolutional | 96x10x10 | stride = 2x2 | 96x60x60 |
| convolutional | 96x6x6 | stride = 2x2 | 96x28x28 |
| maxpooling | 2x2 | stride = 2x2 | 96x14x14 |
| convolutional | 128x3x3 | stride = 1x1 | 128x12x12 |
| convolutional | 160x3x3 | stride = 1x1 | 160x10x10 |
| convolutional | 192x3x3 | stride = 1x1 | 192x8x8 |
| maxpooling | 2x2 | stride = 2x2 | 192x4x4 |
| flatten | | | (3072+$M$)x1 |
| fully-connected | 512x3072 | | 512x1 |
| dropout | 512x1 | $p = 0.5$ | 512x1 |
| fully-connected | 512x512 | | 512x1 |
| dropout | 512x1 | $p = 0.5$ | 512x1 |
| fully-connected | $n$x512 | | $n$x1 |

CNN256 is a scaled-up version of CNN128 for input images with higher spatial resolution. The main difference between the two architectures is that CNN128 takes in 128x128 sized images and CNN256 takes in 256x256 sized images as the input.
CNN256 consists of approx. four million trainable parameters whereas CNN128 consists of approx. three million trainable parameters. As a comparison, the AlexNet architecture consists of approx. 60 million trainable parameters (Krizhevsky et al., 2012).





**Table 4.** The sequential structure of CNN256. The network consists of roughly four million trainable parameters. The number of parameters depends on the number of classes.

| Layer | Layer dimensions | Parameters | Output dimensions |
|---|---|---|---|
| input | 256x256 | | 256x256 |
| convolutional | 96x10x10 | stride = 2x2 | 96x124x124 |
| convolutional | 96x6x6 | stride = 2x2 | 96x60x60 |
| maxpooling | 2x2 | stride = 2x2 | 96x30x30 |
| convolutional | 128x4x4 | stride = 1x1 | 128x27x27 |
| convolutional | 160x4x4 | stride = 1x1 | 160x24x24 |
| maxpooling | 2x2 | stride = 2x2 | 160x12x12 |
| convolutional | 192x3x3 | stride = 1x1 | 192x10x10 |
| maxpooling | 2x2 | stride = 2x2 | 192x5x5 |
| flatten | | | 4800x1 |
| fully-connected | 512x4800 | | 512x1 |
| dropout | 512x1 | $p = 0.5$ | 512x1 |
| fully-connected | 512x512 | | 512x1 |
| dropout | 512x1 | $p = 0.5$ | 512x1 |
| fully-connected | $n$x512 | | $n$x1 |

## 2.3 Description of experiments

CNN128 was trained for 70 epochs on the Subset50 training data with the Nesterov's method (Nesterov, 2013) using the
learning rate of 0.01, momentum of 0.9 and batch size of 256. The parameters are based on small-scale empiric tests where it
was observed that the CNN can be trained successfully with these parameters. The network accuracy was tested every 5 epochs
with the appropriate testing set.

CNN256 was trained for 70 epochs on the Subset100, Subset50, Subset20 and Subset10 training data with the Nesterov's
method using the learning rate of 0.01, momentum of 0.9 and batch size of 256. Additionally, CNN256 was trained on a
variation of Subset50 where the images that were too small for the network input were only padded, instead of having them
enlarged first and then having them padded.

The CNNs were compared with a classifier based on a Random Decision Forest (RDF) utilizing handcrafted features (Sosik
and Olson, 2007). The RDFs were created from the Subset100, Subset50, Subset20 and Subset10 training data consisting of
1000 decision trees.







### 2.4 Evaluation criteria


Cross validation was used to evaluate the performance. The number of cross validations for each data subset was selected based on the number of images per class. Classifiers trained with the data subsets containing less testing images per class were cross validated more in order to have more confidence on the result. The results for CNN128 were cross validated 30 times, whereas the results for CNN256 for Subset100, Subset50, Subset20 and Subset10 were cross validated 30, 30, 30 and 60 times

respectively. To evaluate the performance and to compare different methods, the classification performances were measured using top-$k$ accuracies and confusion matrices. The top-$k$ accuracy is defined as

$$A_k = \frac{T_k}{T_k + F_k} \tag{1}$$

where $T_k$ is the number of test images for which the correct class is within the top-$k$ model predictions, and $F_k$ is the number of images for which the correct class is within the top-$k$ model predictions. The standard deviations were also calculated for

the top-1 accuracies of the cross validations to represent the variation in the performance.

## 3 Results

The overall performance of the classification was good, and the network was able to identify many key species of the Baltic Sea phytoplankton community. Tests with different subsets all reached high performance. A noticeable number of misclassifications occurred between subclasses of close taxonomic affiliation. When the subclasses were combined, CNN256 obtained classifi-

cation accuracies of 0.874 for Subset10, 0.892 for Subset20, 0.889 for Subset50, and 0.887 for Subset100. The classification accuracies for each class with CNN256 on Subset10 with subclasses combined can be seen in Table 5.

The main classification problems occurred evidently with classes having similar resemblance, typically of close taxonomic relation. The highest confusions were within different classes of dinoflagellates and between a species level class and more generic level class belonging to the same order. Table 6 illustrates the most common classification errors. Each row shows

a pair of classes that were commonly confused with each other. It should be noted that the example images in the table are randomly selected and were not necessarily misclassified during the experiments.

As a promising result for the real-time/near real-time method utilization, a relatively small number of trainable parameters in the CNN128 architecture kept the training time short, which made it ideal to test various modifications to the architecture and plankton classification algorithm. The results with the different variations of CNN128 are shown in Table 7. It can be seen

that using metadata as additional features does not increase the top-1 classification accuracy. Using the mode of the images as the padding color results in higher classification accuracy than white or black as the padding color. Resizing images with the bicubic interpolation also results in a better top-1 accuracy than the nearest-neighbor interpolation. A small increase in the top-1 accuracy of CNN128 is observed when using the parametric ReLU activation function.

A small decrease in the top-1 accuracy of the network is observed when opting not to use cropping as a data augmentation

method. A significant decrease in the network top-1 accuracy can be seen when no data augmentation methods are used. The importance of using dropout can also be seen. The top-1 accuracy of the network drops drastically when opting not to use





**Table 5.** The classes in Subset10, their total number of images and classification accuracies with CNN256 without subclasses.

| Class label | Images | Accuracy | Class label | Images | Accuracy |
|---|---|---|---|---|---|
| *Monoraphidium contortum* | 19 | 0.99 | *Teleaulax* sp. | 413 | 0.84 |
| *Skeletonema marinoi* | 756 | 0.99 | *Aphanothece paralleliformis* | 29 | 0.83 |
| Cyst | 150 | 0.97 | *Licmophora* sp. | 62 | 0.83 |
| Calibration bead | 100 | 0.96 | *Heterocapsa triquetra* | 1433 | 0.82 |
| *Heterocapsa rotundata* | 569 | 0.96 | *Chaetoceros similis* | 24 | 0.81 |
| *Peridiniella catenata* single cell | 871 | 0.96 | Ciliata | 201 | 0.8 |
| *Pseudopedinella* sp. | 829 | 0.96 | *Chaetoceros* sp. | 916 | 0.79 |
| *Snowella / Woronichinia* sp. | 3710 | 0.95 | *Eutreptiella* sp. | 190 | 0.78 |
| Chroococcales | 1553 | 0.94 | *Melosira arctica* | 24 | 0.78 |
| *Cymbomonas tetramitiformis* | 150 | 0.94 | *Oocystis* sp. | 458 | 0.77 |
| *Mesodinium rubrum* | 633 | 0.94 | Gymnodiniales | 31 | 0.75 |
| *Pauliella taeniata* | 13 | 0.94 | Cryptomonadales | 138 | 0.73 |
| *Uroglenopsis* sp. | 69 | 0.94 | Cryptophyceae | 175 | 0.73 |
| *Aphanizomenon flosaquae* | 821 | 0.93 | *Cyclotella choctawhatcheeana* | 55 | 0.70 |
| *Dinophysis acuminata* | 73 | 0.93 | Dinophyceae | 2268 | 0.70 |
| Heterocyte | 225 | 0.93 | *Gymnodinium* sp. | 133 | 0.67 |
| *Peridiniella catenata* chain | 140 | 0.93 | *Nodularia spumigena* | 80 | 0.65 |
| *Ceratoneis closterium* | 33 | 0.92 | Oscillatoriales | 15 | 0.64 |
| *Thalassiosira levanderi* | 650 | 0.92 | *Scenedesmus* sp. | 13 | 0.64 |
| *Prorocentrum cordatum* | 230 | 0.91 | *Amylax triacantha* | 21 | 0.62 |
| Pennales | 198 | 0.90 | Akinete | 23 | 0.59 |
| *Pyramimonas* sp. | 623 | 0.89 | *Binuclearia lauterbornii* | 13 | 0.59 |
| Unidentified species 1 | 72 | 0.88 | *Merismopedia* sp. | 138 | 0.58 |
| *Dolichospermum / Anabaenopsis* sp. | 1223 | 0.87 | Euglenophyceae | 42 | 0.56 |
| *Katablepharis remigera* | 85 | 0.87 | Chlorococcales | 88 | 0.48 |
| *Chaetoceros throndsenii* | 12 | 0.86 | *Apedinella radians* | 10 | 0.34 |
| Centrales | 249 | 0.85 | | | |

dropout. Having 1024 nodes in the fully-connected layers resulted in a 0.804 ± 0.021 top-1 accuracy at 70 epochs, the network was then trained to 100 epochs and the top-1 accuracy increased to 0.813 ± 0.018. Increasing the number of nodes at the fully connected layers has clear potential to increase the top-1 accuracies. The increase from 512 nodes to 1028 nodes roughly 205 doubled the number of parameters in the network and resulted in a considerably longer training time. Removing the last pooling layer of the network clearly reduced the network top-1 accuracy and also roughly quadrupled the number of parameters in the



**Table 6.** Class pairs with the highest number of inter-class classification errors in Subset10. The table contains each class (Class A) with higher than 10% confusion to another class, as well as the class with the highest confusion (Class B). The middle column contains the portions of images in Class A that were incorrectly classified to Class B.

| Class label A | | A→B (%) | | Class label B |
|---|---|---|---|---|
| *Apedinella radians* | | 38 | | *Pseudopedinella* sp. |
| Chlorococcales | | 25 | | *Snowella / Woronichinia* sp. |
| *Amylax triacantha* | | 22.5 | | *Peridiniella catenata* single cell |
| *Nodularia spumigena* | | 22.5 | | *Aphanizomenon flosaquae* |
| *Gymnodinium* sp. | | 22 | | Dinophyceae |
| *Merismopedia* sp. | | 20 | | *Snowella / Woronichinia* sp. |
| Euglenophyceae | | 17 | | Cryptomonadales |
| Dinophyceae | | 16 | | *Heterocapsa triquetra* |
| Oscillatoriales | | 15.8 | | *Aphanizomenon flosaquae* |
| *Aphanothece paralleliformis* | | 15 | | Chroococcales |
| Cryptophyceae | | 15 | | *Teleaulax* sp. |
| *Heterocapsa triquetra* | | 12.5 | | Dinophyceae |
| *Scenedesmus* sp. | | 10 | | Chroococcales |





**Table 7.** The top-1 classification accuracies with the variations of CNN128 on Subset50.

| Variation | The top-1 accuracy |
|---|---|
| No variation | 0.809± 0.014 |
| Original image dimensions as features | 0.807± 0.017 |
| Original image dimensions, average and peak photomultiplier tube values as features | 0.809± 0.018 |
| White as a padding color | 0.801± 0.016 |
| Black as a padding color | 0.793± 0.014 |
| Resize images with the nearest-neighbor interpolation | 0.804± 0.018 |
| Parametric ReLU | 0.812± 0.017 |
| No cropping as an augmentation method | 0.800± 0.015 |
| No data augmentation | 0.713± 0.014 |
| No dropout layers | 0.727± 0.026 |
| 1024 nodes in fully-connected layers | 0.804± 0.021 |
| Last pooling layer removed | 0.793± 0.018 |

network. The final experiments were performed using a larger CNN256 network that was tuned based on the observations above.

A comparison of the top-1 accuracies between CNN256 and the Random Forest implementation can be seen in Table 8.
The results of the Random Forest implementation for the data subsets Subset100, Subset50, Subset20 and Subset10 were cross validated 10, 10, 30 and 60 times respectively. The results show that the CNN performs significantly better than the Random Forest implementation with the same data.

**Table 8.** The top-1 classification accuracies of CNN256 and Random Forest (RF), with the different data subsets and with all subclasses included as separate classes.

| Data subset | CNN256 | RF |
|---|---|---|
| Subset100 | 0.848± 0.015 | 0.701± 0.007 |
| Subset50 | 0.827± 0.014 | 0.632± 0.016 |
| Subset20 | 0.792± 0.022 | 0.556± 0.024 |
| Subset10 | 0.782± 0.038 | 0.563± 0.031 |





## 4   Discussion

The Baltic Sea phytoplankton community is a unique mixture of species from both marine and freshwater environments,
that have been adapted to different conditions along the salinity gradient. The community follows a seasonal pattern char-
acterized by a dominance of dinoflagellates and diatoms in spring, after which the community shifts to an early summer
biomass minimum with e.g. different types of flagellates. Along the summer, the community most often becomes dominated
by nitrogen-fixing cyanobacteria, sometimes causing high-magnitude blooms. Different flagellates and other types of small-
sized prokaryotes are a common feature of the community across seasons.

Extensive long-term monitoring records exist from the Baltic Sea as it has long been subject to anthropogenic pressures
leading to eutrophication (Andersen et al., 2017). Therefore, the plankton communities have been studied widely, however
there are many unresolved issues related to phytoplankton ecology. One such issue is the set of factors regulating the spring
bloom phytoplankton community composition (Spilling et al., 2018). Other such issue is the summer cyanobacterial blooms
that are an annual nuisance for the recreational users of the Baltic Sea, and the species composition, timing and magnitude of the
blooms are difficult to predict as the controlling factors are still something of a puzzle (Kahru and Elmgren, 2014; Kownacka
et al., 2018; Kahru et al., 2020). To deepen our understanding of these still partly unresolved processes the utilization of new
methods, such as imaging flow cytometry, plays a key role. There is no perfect way of treating the image data and classification
of all the collected images is not a realistic task as the data will always include images of detritus, other types of unknown
material, and images with multiple objects. However, if the method can be implemented with different taxonomic levels of
identification there is a possibility to discriminate between major taxonomic and functional phytoplankton groups as well as to
acquire even species-level information of multiple classes. Therefore, it is also important to identify the key issues related to
the most commonly confused classes.

Characteristic features tend to lead to better classification of images. Many of the Baltic Sea dinoflagellates do not have
specific features visible on the IFCB images that could be used to distinguish them from one another. Consequently, some of the
highest confusions among the classes were between classes of dinoflagellate taxa. The high confusions within the dinoflagellate
classes are also partly explained by the confused classes being on a higher and lower level of the same taxonomic hierarchy.
For example, the general class Dinophyceae, a diverse class including all unidentified dinoflagellates, easily engulfs images
belonging to the other dinoflagellate classes. There are also other classes in the dataset that are on a higher and lower level
of the same taxonomic branch and are thus understandably confused such as the classes *Aphanothece paralleliformis* and
Chroococcales, where *A. paralleliformis* is a species belonging the order Chroococcales. A similar case is the class *Teleaulax*
sp. which belongs to the class Cryptophyceae. Similarly, *Nodularia spumigena* and Oscillatoriales which both got confused
with *Aphanizomenon flosaquae*, are all classes of filamentous cyanobacteria. High confusions occurred also among classes
that represent different species of the same order, such as *Apedinella radians* and *Pseudopedinella* sp. that both belong to the
same order Pedinellales, as well as the classes *Merismopedia* sp. and *Snowella / Woronichinia* sp. which are of the same order
Synechococcales (Table 6).



The three major summer cyanobacterial bloom-forming taxa in the Baltic Sea are *Nodularia spumigena*, *Aphanizomenon flosaquae* and *Dolichospermum* sp. (Niemistö et al., 1984; Stal et al., 2003; Olofsson et al., 2020) of which only *A. flosaquae* is not potentially toxic. Separating the three classes is thus of high importance, and it was already achieved with the class *Dolichospermum* sp. The class *A. flosaquae* was also identified with high accuracy but the problem was false positive classifi-

cations from both classes *N. spumigena* and Oscillatoriales. The main explanation for these confusions is the small amount of training data of the two classes compared to the similar-looking class of *A. flosaquae* having a much larger training set (Table 5). To address the class confusions, especially those with high ecological relevance, separate pairwise or group-wise classifiers or approaches that force the CNN to find the subtle differences among similar classes could be used. These approaches have been utilized in fine-grained visual classifications for other applications (Dubey et al., 2018; Sun et al., 2019; Li and Monga, 2020).

The CNNs are mostly trained with large training sets consisting at least hundreds but rather thousands of training images per class. This is still often not feasible as the training sets and the classifier usually only work for the community of origin. In our study we included also classes of only 10-50 images per class and still reached a high classification accuracy among some of those classes, such as *Monoraphidium contortum*, *Pauliella taeniata* and *Chaetoceros throndsenii* (Table 5). This was highly affected by the efficient data augmentation methods, which clearly improved the classification accuracy as supported by

the results of e.g. Correa et al. (2017). Another important factor contributing to these classes reaching high accuracy is that all of these classes had very specific shape compared to other classes. Still even in our data it is clear that having more than 100 training images has a great effect on the classification accuracy (Table 7).

The CNN-based method outperformed the Random Forest-based method by a large margin (Table 8). This is most likely due to the deep feature extraction being superior to traditional feature extraction methods that require manual engineering.

This result agrees with previous studies (e.g. Keçeli et al., 2017). Including image size as an additional feature did not improve the classification accuracy. For a phytoplankton specialist, the size of the organism is in many cases one of the key pieces of information for species identification. This surprising result is similar to those obtained by Ellen et al. (2019). However, this requires more research. Including metadata as features to CNN is not straightforward and the used approach heavily affects the network's ability to utilize the additional features in the classification. Recently more sophisticated approaches have been

proposed and shown to provide benefit to other classification tasks such as fine-grained classification of animal species using geolocation (Chu et al., 2019).

One interesting issue is why CNNs are not yet more widely utilized for classifying and analyzing "real" ecological plankton datasets. While the deep feature extraction outperforms handcrafted features, also the handcrafted features perform well for several phytoplankton groups classified originally with both SVM and RDF-based classifiers (Sosik and Olson, 2007; Laney

and Sosik, 2014; Anglès et al., 2015, 2019; Bueno et al., 2017). At the moment, most ecological studies using phytoplankton datasets collected with an IFCB base the classification on the features and method developed by Sosik and Olson (2007), but using RDF instead of SVM. One reason for this is that the typical CNNs struggle in open-class problems where the method is applied to novel data with classes not present in the training data (e.g. new species or detritus and dirt). This is mainly due to the softmax layer in the network architecture to convert neuron activations into class probabilities. This forces the

network to output a relatively high probability to certain class from the training set even if the input image is from a novel class





which makes it difficult to identify when the network cannot recognize the image. One way to address this is to utilize metric learning (Mensink et al., 2012) or to use additional filtering process after the classification by Luo et al. (2018).

Another issue related to real-life utilization of CNNs is that they often require a notable amount of computation resources especially if they are trained from scratch. In this study we utilized the AlexNet (Krizhevsky et al., 2012) as a base but the

CNN architecture was modified to meet the requirements of a lighter structure suitable for repetition of tests. We showed that it is possible to obtain a good accuracy with a relatively shallow architecture and small amount of training data. Many of the general image classification architectures like AlexNet (Krizhevsky et al., 2012), VGGNet (Simonyan and Zisserman, 2014), ResNet (He et al., 2016) and GoogLeNet (Szegedy et al., 2015) can be utilized and fine-tuned for plankton image classification tasks with great performance (Dai et al., 2016b; Li and Cui, 2016; Orenstein and Beijbom, 2017; Pedraza et al., 2017). Their

advantage is their versatility and availability of pre-trained models that can be tuned for plankton recognition with a limited amount of training data using transfer learning, but as shown for ZooplanktoNet, a CNN architecture specifically developed for zooplankton classification outperformed general-purpose image classification architectures in the studied data set (Dai et al., 2016a). However, building and training a CNN for each plankton data separately is not realistic due to the lack of expertise of machine learning and availability of the computing resources among plankton researchers. Therefore, there exists a need for a

model library with pre-trained CNN models for different plankton communities. This would make it possible for the researches to obtain a pre-trained model with the most similar class composition and fine-tune it for their data with moderate amount of training data and reasonable computing resources.

## 5   Conclusions

Multiple studies have shown CNNs as a functional tool in classification tasks of plankton. However, real-life utilization of

CNNs for plankton image datasets is still scarce, especially in the case of microplankton, and RDFs are still more often in operational use. High confusions are often related to close taxonomic affiliations, which is not always an issue if the interest is not in the species-specific dynamics but rather in the dynamics of larger functional groups, for example. However, one challenge to overcome with CNNs is the open-class problem. When having plankton time series collected with an imaging flow cytometer, it is naive to assume that all of the collected images could be assigned to a specific class. It is impossible to

create classes for all images of small roundish or elongated objects, the multiple appearances of detritus and other objects that are impossible to identify and, therefore, it is necessary to find a solution to the open-class problem before broader operational use can be achieved, because it is impossible to make training sets for all possible new classes ahead and they need to be filtered out from the classification results as well as detritus and sort. There are also other issues to be resolved before the operational use of CNNs is possible, one of them being the computing resources available for researchers. We showed, however, that it

should not become a problem as it is also possible to reach a high classification accuracy with a rather shallow CNN architecture requiring less computing resources. Another issue related to the operational use, not limited to only CNNs, is the need for large training sets. It can be overcome by using suitable data augmentation methods as shown, but there still remains significant potential in having larger training sets. Both open image and model libraries with pre-trained CNN models and training images





for different plankton communities would be beneficial and would accelerate the exploration of the vast amount of plankton

datasets already collected among the multitude of monitoring programs and research projects around the world.

*Author contributions.* All authors contributed to the design and development of the work. The experiments were carried out by TE and OG. The analysis of the obtained results was carried out by TE, OG, and KK. Expert labeling of the image data for model training and evaluation purposes was carried out by KK. All authors contributed in writing the paper.

*Competing interests.* The authors declare that they have no conflict of interest.

*Acknowledgements.* The research was carried out in the FASTVISION project (No. 321980 and 321991) funded by the Academy of Finland and it was partially funded by H2020-project JERICO-NEXT (grant agreement no. 654410). The work was also partly written by the personal grant of Kaisa Kraft from Tiina and Antti Herlin foundation.



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
