# Peer review of "Towards operational phytoplankton recognition with automated high-throughput imaging and compact convolutional neural networks"

_Ocean Science, 2020_

## Referee Comment (RC1) · Anonymous Referee #1 · 4 Aug 2020

GENERAL COMMENTS

The manuscript addresses automated classification of brackish-water phytoplankton images collected in the Baltic Sea. Images used in the experimental part were collected using the IFCB (Imaging FlowCytobot) system.

There is an extensive discussion regarding current state and challenges around plankton image classification, including issues such as the need of Computer Vision and Machine learning based approaches to deal with the large volume of data, limitations

of traditional machine learning based approaches that require hand-designed feature engineering and therefore do not scale well from one case to another, recent advances towards deep learning based approaches as they are able to automatically extract features from raw images, the need of large amounts of training data, imbalanced class sizes and the need to deal with open classes or unconstrained class (e.g., detritus), and so on. There is also a clear concern with respect to practical implications of using these classifiers.

Effectively, a reduced scale size CNN network is proposed and trained from scratch, varying some training conditions, for datasets in the above Baltic Sea context. Classification accuracy up to 0.85 is achieved and observed confusion cases are extensively discussed. The proposed network and performed experiments are apparently meant to show that a small scale CNN can be trained from scratch on limited and imbalanced training data condition, using data augmentation strategies. This agrees with what is stated in the Abstract: "Our results show that it is possible to obtain good classification accuracy with relatively shallow architectures and a small amount of training data when using effective data augmentation methods even with a very unbalanced dataset."

These issues, training small networks from scratch, using data augmentation, dealing with imbalanced classes, have been already at some extent addressed in previous publications on plankton image classification. Motivating issues as well as discussions regarding challenges of making effective practical use of these classification methods are also discussed in previously published papers. Since no comparison is provided, it is not clear what are the contributions of the presented method and experimental results.

In summary, the objectives and the contributions of the study are not clear.

SPECIFIC COMMENTS

It is stated that "Our approach ... is to address some fundamental challenges in phytoplankton identification". What are precisely these fundamental challenges?

[Figure]

A convolutional neural network architecture is proposed, trained, and results are presented. Some results refer to input of size 128x128 while others to input of size 256x256. However there is no comparison between them nor with any other existing models or datasets. This makes very difficult to evaluate the relevance of the study. In particular, since there are many lightweight models being proposed for image classification tasks in general, one wonders why none was considered or at least used as a reference.

There are several reports, not only with respect to plankton images, of how transfer learning (models pre-trained on images of a distinct domain and fine-tuned with target domain images) often generates classifiers that achieve good classification rates. It seems important to compare the results achieved with the proposed network and results that could be achieved by fine-tuning pre-trained models. Note that models pre-trained on ImageNet data are often used, but any model could be also pre-trained on any large datasets (e.g., on large plankton datasets).

Dataset description in the "Materials and methods" section is confusing. As stated, and as listed in Table 1, it consists of 53 classes. Then it is said that they are further subdivided in subclasses that results in a total of 61 classes. Where or how this subdivided case is explored ?

Experimental setup: The first main issue is what the experiments are trying to convey (with respect to current state-of-the-art). Then, there are some details related to experimental setting that needs clarification: (1) if evaluation is performed based on cross-validation, why the dataset was separated into training-testing sets (25% for testing) ? (2) When performing cross-validation, did you consider stratified folds? (3) It is stated that "The parameters are based on small-scale empiric tests where it was observed that the CNN can be trained successfully with these parameters." What kind of empirical tests are they? Did you take care so as to not introduce any bias (parameters chosen on privileged information) ? (4) When comparing CNN with RF, it seems that different cross-validation fold numbers have been considered. To avoid performance

differences that may be due to fold differences, the same folds should be considered for training/testing of both algorithms. (5) In general, establishing a baseline case helps comparison; for instance, as results are presented, it is not clear why some results refer to CNN128 and others to CNN256. How they compare each other?

TECHNICAL / SMALL DETAILS

"collaboration between experts and exchange with other disciplines, like modelers". I did not understand what is the meaning of "modeler" here.

"The number of images in a subset assigned to the testing set is equal to 25% of the threshold value of the subset. The remaining images, up to one thousand images, are then assigned to the training set." Do you mean "up to one thousand images PER CLASS" ?

Table 2: does the number of test images per class refer to the smallest class? If so, separation of 25% for testing was class-wise ? (but still, it is not clear where this division is considered)

Data augmentation: with 90-degree rotation, we have 0-degree (original), 90-degree, 180-degree, and 270-degree. So it would be a 3x augmentation and not 4x, right ?

"Validating any results related to class specific classification accuracies becomes computationally expensive since there are few images in a moderate number of classes". This sentence is confusing for me. If there are many images and large number of classes, I would understand that evaluation may become computationally expensive, but the other way is not clear.

Table 3 and Table 4: description of the architecture is not following the standard. For instance, when specifying the dimensions of a tensor, the standard is h x w x d (height x width x depth). Also, since you are using non-usual filter sizes (masks 10x10, 6x6, 4x4), their choice should be justified.

Batch size is a hyperparameter that can greatly affect convergence. There are some

recommendations to use relatively small batch sizes (32 or 64) or to reduce it along the epochs. It could be interesting to evaluate different values.

Which loss function was used? Did you use any kind of regularization other than dropout ?

The first time I read the part that mentions cross-validation, I have understood 30,30,30, 60 were referring to the number of folds. But at a second reading, it seems to refer to the number of repetitions. Could you precisely describe how you did each cross-validation ?

Since computational cost is a concerning issue, which type of processor has been used and how long was the training time ? What is considered a "short training time" ?

Would it be possible to display results in a confusion matrix ? At least the more interesting cases ? It is difficult to follow the results in Tables, as shown.

Image size is mentioned as a metadata used in the experiments. If properly cropped, image size could reflect organism size in terms of pixels. However, for this information to be useful, size should correspond to the physical size of the organism. Estimating the actual physical size may not be so simple if we do not have precise distance information.

"typical CNNs struggle in open-class problems where the method is applied to novel data with classes not present in the training data". Is this not true for most of the machine learning algorithms ?

---

## Referee Comment (RC2) · Anonymous Referee #2 · 14 Aug 2020

The paper "Towards operational phytoplankton recognition with automated high-throughput imaging and compact convolutional neural networks2"presents how a conventional neural network method can be used to automatically recognise phytoplankton classes in real time (or near real time) in the Baltic Sea. This approach uses images collected by an in situ optical sensor. The study is in the scope of the special issue of JERICO-RI where the biological component has been investigated using different techniques such as the image recognition using different bio-optical sensors. It is a very clear, and easy to understand even without deep background in mathematical

processes and operations. One of the main limitations to use the Deep learning approach is the need of a large number of images and equal quantity per group to build the training sets. Therefore, the authors explore different techniques to increase the performance of the approach. I recommend the publication. However, the article will gain in mentioning also: 1) the current efforts of imaging data management with the establishment of current international biodiversity data standards, such as Darwin Core (DwC), used OBIS (EUROBIS) and GBIF. 2) what are the benefits to use the approach in the article versus the established tools such as ECOTAXA based on Deep Machine Learning also.

Comments:

Line 62: "FlowCytobot is among the most frequently used imaging flow cytometer". Until now, there was only one or two groups of American scientists using the FlowCytobot. I do not think that we can say most frequently used in this case.

Line 80: what are the practical implications to aquatic research which are mentioned? This needs clarification.

Line 100: it will be worthwhile here to mention the principal of a FAIR data: findability, accessibility, interoperability, and reusability.

Line 109: "FerryBox"

Line126: is it testing set or training set is equal to 25% Needs clarification (see Table2)? Why 25% has been chosen

Line 211: "CNN performs significantly better than the Random Forest implementation". It should be mentioned that that the two methods used different attributes with a higher number used for CNN which explains a better performance for CNN.

Line 226: in what identifying the planktonic species is important for the Baltic Sea ecosystem? I understood that the authors want to relate their mathematical approach to an ecological interest but it will be relevant to have some information about why

monitoring the species is important, particularly for those who do not know the Baltic Sea.

Line 253: "ecological relevance" should be better to mention human health concern?

Line 294: "there exists", replace by there is.

Line 304: " It is impossible to create classes for all images. . .." This sentence underlines the lack of information concerning the percentage of phytoplankton recognised compared to the on those which are not recognised and potentially included in " small roundish or elongated objects".

---

## Author Comment (AC1) · 2 Sep 2020

C1: These issues, training small networks from scratch, using data augmentation, dealing with imbalanced classes, have been already at some extent addressed in previous publications on plankton image classification. Motivating issues as well as discussions regarding challenges of making effective practical use of these classification methods are also discussed in previously published papers. Since no comparison is provided, it is not clear what are the contributions of the presented method and experimental results

[Figure]

A: Thank you for the valuable comments. It should be noted that we are not trying to claim that the machine learning approaches we have used in this paper are unprecedented, but we are applying them into a dataset collected from a completely new type of habitat with a species composition different from the previous studies. This creates possible challenges for the algorithms and modifications to the classification method may be required. We are not trying to fundamentally compare and discriminate between the best technical solutions (in that case this paper would have been targeted elsewhere) but how those technical solutions reflect to operational utilization of the CNNs in phytoplankton recognition. The contributions will be further clarified in revised version.

C2: It is stated that "Our approach ... is to address some fundamental challenges in phyto- plankton identification". What are precisely these fundamental challenges?

A: The fundamental challenges are as follows: 1) Large class imbalance: it is easier to obtain huge set of training images from typical species, but training sets of many classes are difficult to extend. 2) Size of the CNN architecture: not many marine biologists have the access to high computing resources. These will be further clarified in the revised version of the paper.

C3: A convolutional neural network architecture is proposed, trained, and results are presented. Some results refer to input of size 128x128 while others to input of size 256x256. However there is no comparison between them nor with any other existing models or datasets. This makes very difficult to evaluate the relevance of the study. In particular, since there are many lightweight models being proposed for image classification tasks in general, one wonders why none was considered or at least used as a reference.

A: The experiments on both CNN128 and CNN256 were carried out using the same datasets and therefore, the classification accuracies (e.g. 0.809 vs. 0.827 for Subset50) are comparable. A more direct comparison between these two architectures

will be included to the revised version. It should be also noted that relevance of the study should not depend on the comparison of numbers but how it is connected to the context it is referring to.

C4: There are several reports, not only with respect to plankton images, of how transfer learning (models pre-trained on images of a distinct domain and fine-tuned with target domain images) often generates classifiers that achieve good classification rates. It seems important to compare the results achieved with the proposed network and results that could be achieved by fine-tuning pre-trained models. Note that models pre-trained on ImageNet data are often used, but any model could be also pre-trained on any large datasets (e.g., on large plankton datasets).

A: One reason to utilize shallower architectures is to allow the training from scratch with a limited amount of training data and this way avoid a computationally heavy pre-training process.

C5: Dataset description in the "Materials and methods" section is confusing. As stated, and as listed in Table 1, it consists of 53 classes. Then it is said that they are further subdivided in subclasses that results in a total of 61 classes. Where or how this subdivided case is explored?

A: Classifiers were trained on the full set of 61 classes and for the final evaluation results the subclasses were combined. This will be further clarified in the revised version.

C6: Experimental setup: The first main issue is what the experiments are trying to convey (with respect to current state-of-the-art). Then, there are some details related to experimental setting that needs clarification: (1) if evaluation is performed based on cross-validation, why the dataset was separated into training-testing sets (25% for testing)? (2) When performing cross-validation, did you consider stratified folds? (3) It is stated that "The parameters are based on small-scale empiric tests where it was observed that the CNN can be trained successfully with these parameters." What kind of empirical tests are they? Did you take care so as to not introduce any bias (parameters

chosen on privileged information) ? (4) When comparing CNN with RF, it seems that different cross-validation fold numbers have been considered. To avoid performance differences that may be due to fold differences, the same folds should be considered for training/testing of both algorithms. (5) In general, establishing a baseline case helps comparison; for instance, as results are presented, it is not clear why some results refer to CNN128 and others to CNN256. How they compare each other?

A: (1-2) the evaluation was done using repeated random subsampling cross-validation, i.e., training was repeated N times with randomly selected training and test sets. (3) Preliminary tests were carried out to find out such hyperparameters that the CNN model converges during the training. The classification accuracies very not used to optimize these parameters. (4) Since the random subsampling validation was used, the number of repetitions does not have major effect on the results as long as the amount of repetitions is large enough. The larger amount of repetitions results in more reliable results. (5) CNN256 outperforms CNN128. A more direct comparison between these two architectures will be included to the revised version.

C7: "collaboration between experts and exchange with other disciplines, like modelers". I did not understand what is the meaning of "modeler" here.

A: Modelers are scientists who are developing models that are used in for example predicting or understanding harmful algal blooms.

C8: "The number of images in a subset assigned to the testing set is equal to 25% of the threshold value of the subset. The remaining images, up to one thousand images, are then assigned to the training set." Do you mean "up to one thousand images PER CLASS" ?

A: Yes, on thousand images per class. Thank you for the correction.

C9: Table 2: does the number of test images per class refer to the smallest class? If so, separation of 25% for testing was class-wise ? (but still, it is not clear where this

division is considered) Data augmentation: with 90-degree rotation, we have 0-degree (original), 90-degree, 180-degree, and 270-degree. So it would be a 3x augmentation and not 4x, right?

A: Yes, the amount of the rotation augmented images is 3 times the number of original images, so after augmentation the total number of images is 4 times larger than before the augmentation.

C10: "Validating any results related to class specific classification accuracies becomes computationally expensive since there are few images in a moderate number of classes". This sentence is confusing for me. If there are many images and large number of classes, I would understand that evaluation may become computationally expensive, but the other way is not clear.

A: The sentence refers to the computationally expensive nature of the repeated random subsampling validation. This will be clarified in the revised version of the paper.

C11: Table 3 and Table 4: description of the architecture is not following the standard. For instance, when specifying the dimensions of a tensor, the standard is h x w x d (height x width x depth). Also, since you are using non-usual filter sizes (masks 10x10, 6x6, 4x4), their choice should be justified.

A: Thank you for the comments. This will be fixed in the revised version.

C12: Batch size is a hyperparameter that can greatly affect convergence. There are some recommendations to use relatively small batch sizes (32 or 64) or to reduce it along the epochs. It could be interesting to evaluate different values.

A: Thank you for the comment. We will consider evaluating this.

C13: Which loss function was used? Did you use any kind of regularization other than dropout?

A: The categorical cross entropy was used as the loss function. No other types of

regularization were used in addition to dropout.

C14: The first time I read the part that mentions cross-validation, I have understood 30,30,30, 60 were referring to the number of folds. But at a second reading, it seems to refer to the number of repetitions. Could you precisely describe how you did each cross-validation?

A: Validation was done using repeated random subsampling validation (Monte Carlo cross-validation) instead k-fold cross-validation. This will be clarified in the verified manuscript.

C15: Since computational cost is a concerning issue, which type of processor has been used and how long was the training time ? What is considered a "short training time"?

A: We will provide the information about the computer in the revised version. Short computation is, of course, relative and depends on the available computer resources. However, it should be noted that the environmental scientists analyzing the image data typically do not have access to efficient computational resources, therefore, shallower architectures are preferred.

C16: Would it be possible to display results in a confusion matrix ? At least the more interesting cases ? It is difficult to follow the results in Tables, as shown.

A: We generated a confusion matrix first, but decided to the select the current representation as it made it easier to see the visual similarities and differences in the classes that were confused. However, the confusion matrix can be added to the revised version.

C17: Image size is mentioned as a metadata used in the experiments. If properly cropped, image size could reflect organism size in terms of pixels. However, for this information to be useful, size should correspond to the physical size of the organism. Estimating the actual physical size may not be so simple if we do not have precise
distance information.

A: The size variation in plankton images is extreme (from tens of pixels to thousands of pixels). Therefore, scaling is necessary, and it is challenging to preserve the size information in the images.

C18: "typical CNNs struggle in open-class problems where the method is applied to novel data with classes not present in the training data". Is this not true for most of the machine learning algorithms?

A: Yes, for some degree this is true for most classification methods. However, certain classifiers (e.g. statistical classifiers) are better than CNN for identifying when the classifier is not able to recognize the. This is due to the CNN's (softmax) tendency to give relatively high probabilities even if the image is from an unseen class.

---

## Author Comment (AC2) · 2 Sep 2020

C1: Line 62: "FlowCytobot is among the most frequently used imaging flow cytometer". Until now, there was only one or two groups of American scientists using the FlowCytobot. I do not think that we can say most frequently used in this case.

A: Thank you for the comments. We are not trying to say IFCB is the most frequently used, but that it is among the most frequently used imaging flow cytometers. Imaging flow cytometry has only in recent years (ten years or so) emerged as an attractive

method (improved image quality and operationality) for phytoplankton research. There are not many commercially available imaging flow cytometers that are suitable for phytoplankton research and these are FlowCAM, IFCB and Imagestream (e.g. Dashkova et al. 2016, Lombard et al. 2019). We will clarify this in the revised version.

C2: Line 80: what are the practical implications to aquatic research which are mentioned? This needs clarification.

A: Typically studies that are dealing with plankton classification are addressing solely the classification performance of nice, identifiable images. This is possible in the image datasets that are meant for testing and developing machine learning algorithms but is never the case when trying to classify "real ecological datasets". We are referring with "practical implications" to the analysis of the confused classes and considering the classification process from the operational point of view.

C3: Line 100: it will be worthwhile here to mention the principal of a FAIR data: findability, accessibility, interoperability, and reusability.

A: Thank you for correction.

C4: Line 109: "FerryBox"

A: Thank you for correction.

C5: Line 126: is it testing set or training set is equal to 25% Needs clarification (see Table2)? Why 25% has been chosen

A: Number of test images is the same for each class inside each subset. The number is 25% of the minimum number of images per class. For example, with the subset with all classes with at least 100 images, the number of test images for each class is 25.

C6: Line 211: "CNN performs significantly better than the Random Forest implementation". It should be mentioned that that the two methods used different attributes with a higher number used for CNN which explains a better performance for CNN.

A: Could you clarify what you mean by different (higher number of) attributes? Different image features? While we agree that the higher number of features has effect, we believe that the main reason why CNN outperforms Random Forest is the fact that the features are learned from the data.

C7: Line 226: in what identifying the planktonic species is important for the Baltic Sea ecosystem? I understood that the authors want to relate their mathematical approach to an ecological interest but it will be relevant to have some information about why monitoring the species is important, particularly for those who do not know the Baltic Sea.

A: It has been stated in that "cyanobacteria form massive summer blooms" and that "Baltic Sea suffers from eutrophication". We will further clarify the ecology of the Baltic Sea phytoplankton and the importance of the species monitoring in the revised version.

C8: Line 253: "ecological relevance" should be better to mention human health concern?

A: Actually no. In the Baltic Sea the toxicity of the cyanobacteria is not so much related to human health (although of course this is also important) because there is no for example mussel farming through which the toxins would highly affect to humans. Rather the toxins affect the immediate ecosystem around, and are more of a concern for example for dogs. Of course there is a risk for human health also and the summer blooms are monitored extensively but we do not wish to rule this to address only that aspect of the matter.

C9: Line 294: "there exists", replace by there is.

A: Thank you for correction.

C10: Line 304: " It is impossible to create classes for all images. . .." This sentence underlines the lack of information concerning the percentage of phytoplankton recognised compared to the on those which are not recognised and potentially included in "

small roundish or elongated objects".

A: It is very difficult to assess the portion of the images that cannot be classified separately because the image data collected includes so versatile set of these images. The amount very much depends on the study site, the community composition (different one in different seasons), how much decaying matter exists that still contains enough chlorophyll to trigger for image (other words meaning chlorophyll containing trash), etc that assessing this would require a lot of work and would still not be universal estimate.